# Force-FAK signaling coupling at individual focal adhesions coordinates mechanosensing and microtissue repair

Dennis W. Zhou [1,2,9], Marc A. Fernández-Yagüe [2,3,9], Elijah N. Holland[2,4], Andrés F. García[3], Nicolas S. Castro[1], Eric B. O'Neill[2,3], Jeroen Eyckmans [5,6], Christopher S. Chen [5,6], Jianping Fu [7], David D. Schlaepfer[8] & Andrés J. García [2,3✉]

How adhesive forces are transduced and integrated into biochemical signals at focal adhesions (FAs) is poorly understood. Using cells adhering to deformable micropillar arrays, we demonstrate that traction force and FAK localization as well as traction force and Y397-FAK phosphorylation are linearly coupled at individual FAs on stiff, but not soft, substrates. Similarly, FAK phosphorylation increases linearly with external forces applied to FAs using magnetic beads. This mechanosignaling coupling requires actomyosin contractility, talin-FAK binding, and full-length vinculin that binds talin and actin. Using an in vitro 3D biomimetic wound healing model, we show that force-FAK signaling coupling coordinates cell migration and tissue-scale forces to promote microtissue repair. A simple kinetic binding model of talin-FAK interactions under force can recapitulate the experimental observations. This study provides insights on how talin and vinculin convert forces into FAK signaling events regulating cell migration and tissue repair.

[1] Coulter Department of Biomedical Engineering, Georgia Institute of Technology, Atlanta, GA, USA. [2] Petit Institute for Bioengineering and Bioscience, Georgia Institute of Technology, Atlanta, GA, USA. [3] Woodruff School of Mechanical Engineering, Georgia Institute of Technology, Atlanta, GA, USA. [4] School of Chemical and Biomolecular Engineering, Georgia Institute of Technology, Atlanta, GA, USA. [5] Department of Biomedical Engineering, Boston University, Boston, MA, USA. [6] Wyss Institute for Biologically Inspired Engineering, Harvard University, Boston, MA, USA. [7] Department of Mechanical Engineering, Department of Biomedical Engineering, Department of Cell and Developmental Biology, University of Michigan, Ann Arbor, MI, USA. [8] Moores Cancer Center, Department of Obstetrics, Gynecology, and Reproductive Sciences, University of California, San Diego, La Jolla, CA, USA. [9] These authors contributed equally: Dennis W. Zhou, Marc A. Fernández-Yagüe. ✉email: andres.garcia@me.gatech.edu

Mechanical forces are key regulators of development, health, and disease[1–4]. Focal adhesions (FAs), nanoscale complexes of structural and signaling molecules that link the extracellular matrix (ECM) to the cytoskeleton through integrin receptors, function as principal sites of mechanotransduction[5,6]. Studies with contractility inhibitors[7], deformable substrates[8], and laser tweezers[9] have established that force regulates FA assembly and signaling and identified key molecules in these mechanoresponses[10–14], yet very little is known about how forces are integrated into biochemical signals. Most of our understanding of mechanotransduction comes from population-based (e.g., Western blots) or whole-cell assays (e.g., immunostaining of FAs in cells treated with contractility inhibitors) where the cell is viewed as in a uniform stress state. These analyses provide averaged metrics that may not reveal important relationships at the cell-ECM interface due to the heterogeneity of individual FAs in terms of force and composition. Here, we examined the relationship between force and FA signaling at individual FAs by quantifying the localization and Y397 phosphorylation of focal adhesion kinase (FAK) for cells adhering to micropost-array detectors (mPADs)[15]. FAK is an essential non-receptor tyrosine kinase that transduces crucial signals from FAs to regulate diverse cellular activities including survival, migration, and mechanosensing[16]. FAK is recruited to FAs via its focal adhesion targeting (FAT) domain where it binds the cell membrane and other FA proteins such as talin and paxillin to become catalytically active in a multi-step process[17]. Autoinhibited FAK dimers bind phosphatidylinositol 4,5-bisphosphate (PIP$_2$)-rich membranes to disrupt the autoinhibitory interaction between the four-point-one protein, ezrin, radixin, moesin (FERM) and kinase domains and expose the autophosphorylation site tyrosine 397 (Y397) for trans-autophosphorylation. Once phosphorylated, FAK functions as a molecular scaffold to recruit Src kinases to phosphorylate FAK on tyrosines 576 and 577 and become catalytically active. The autophosphorylation of FAK on Y397 is a critical early step in adhesion signaling[18,19]. We show that traction force and FAK localization, as well as traction force and Y397-FAK phosphorylation, are linearly coupled at individual FAs on stiff, but not soft, substrates. Furthermore, using magnetic beads to apply external forces to FAs and an in vitro 3D biomimetic wound healing model, we demonstrate that force-FAK signaling coupling coordinates mechanosensing and microtissue repair.

## Results

**Linear Force-FAK signaling coupling at FAs.** Mouse embryonic fibroblasts (MEFs) were cultured overnight on fibronectin-coated mPADs of either soft (5 kPa) or stiff (14 kPa) elastic modulus. A significant advantage of mPADs over traction force microscopy methods using deformable bulk gels is that the deflection for a particular post is independent of deflections of other posts, allowing isolation of forces and signaling events to the FA of interest. Because cell fixation can affect post deflections, we developed an optimized fixation protocol that preserves traction forces for cells on mPADs (Supplementary Fig. 1). Cells on mPADs were fixed and immunostained for total FAK (tFAK) and FAK phosphorylated at Y397 (pY397-FAK) at FAs (Fig. 1a–e, Supplementary Fig. 2). The integrated intensity (sum of intensities over the FA) for tFAK and pY397-FAK was quantified by confocal microscopy at individual FAs adhering to microposts with known deflections. In this fashion, we constructed spatial heat maps of force and tFAK (Fig. 1f, h) and pY397-FAK (Fig. 1g, i) integrated intensity for individual FAs on soft and stiff mPADs. Cells generate lower average traction forces at FAs on soft substrates (Fig. 1j) compared to stiff substrates ($P < 0.0001$, Fig. 1m). Treatment with the ROCK inhibitor Y-27632 to impair myosin contractility significantly reduces traction forces on both soft ($P < 0.0001$, Fig. 1j) and stiff ($P < 0.0001$, Fig. 1m) substrates, whereas

treatment with the FAK kinase inhibitor, PF-228, has no effect on traction forces on either soft ($P = 0.4013$, Fig. 1j) or stiff ($P > 0.9999$, Fig. 1m) substrates. No differences in average levels of tFAK ($P = 0.6526$, Fig. 1k, n) or pY397-FAK ($P = 0.1655$, Fig. 1l, o) at FAs were observed between soft and stiff substrates, but treatment with either Y-27632 or PF-228 reduced both tFAK (soft: Y-27632 $P = 0.0032$, PF-228 $P = 0.0195$, Fig. 1k; stiff: Y-27632 $P < 0.0001$, PF-288 $P = 0.0004$, Fig. 1n) and pY397-FAK (soft: Y-27632 $P = 0.0002$, PF-228 $P < 0.0001$, Fig. 1l; stiff: Y-27632 $P < 0.0001$, PF-228 $P = 0.0146$, Fig. 1o) levels at FAs compared to controls. These results are consistent with reports that cells generate higher traction forces, which are dependent on actomyosin contractility but not FAK kinase activity[15,20], on stiff compared to soft substrates and that inhibition of actomyosin contractility reduces FAK localization and Y397 phosphorylation at FAs[12,20,21]. Notably, the spatial heat maps for cells adhering to stiff mPADs show that FAs with high forces also have high levels of tFAK and pY397-FAK (Fig. 1h, i). We, therefore, plotted traction force vs. tFAK (Fig. 1p, r) and traction force vs. pY397-FAK (Fig. 1q, s) for individual FAs for soft and stiff mPADs. Strikingly, tFAK and pY397-FAK levels both increase linearly with traction force at FAs for cells on stiff substrates (linear regression force-tFAK $P < 0.0001$; force-pY397-FAK $P < 0.0001$, Fig. 1r, s), but there is no relationship between force and tFAK ($P = 0.1230$) or force and pY397-FAK ($P = 0.1072$) on soft substrates (Fig. 1p, q). Treatment with either Y-27632 or PF-228 disrupts the linear relationship on stiff substrates for force-tFAK (Fig. 1r, Y-27632 $P = 0.8896$, PF-228 $P = 0.2084$) and force-pY397-FAK (Fig. 1s, Y-27632 $P = 0.7986$, PF-228 $P = 0.2260$), indicating that actomyosin contractility and FAK catalytic activity are required for linear force-FAK signaling coupling at FAs. The linear relationships for force-tFAK and force-pY397-FAK were observed across multiple cells (Supplementary Fig. 3), as well as in human mesenchymal stem cells (Supplementary Fig. 4). In addition, fibroblasts adhering to 25 kPa mPADs also display linear force-FAK signaling coupling (Supplementary Fig. 5). Furthermore, treatment with the Src-family kinase inhibitor PP2 does not alter traction forces compared to controls ($P = 0.2541$) and does not disrupt the force-tFAK and force-pY397-FAK linear relationships (Supplementary Fig. 6).

We examined the contributions of talin, an essential FA component regulating integrin receptor activation and connection to the cytoskeleton[22–24], to force-FAK signaling coupling on stiff (14 kPa) mPADs. Treatment with talin-1 shRNA reduces talin-1 levels by 90% and total talin (talin-1 and talin-2) levels by 60% compared to control shRNA (Supplementary Fig. 7). Talin-1 knock-down clearly impacts traction forces, tFAK levels, and pY397-FAK levels at FAs (Fig. 2a–d). Talin-1 depletion reduces total traction force (Fig. 2e, $P = 0.0005$) and average levels of tFAK (Fig. 2f, $P < 0.0001$) and pY397-FAK (Fig. 2g, $P < 0.0001$) at FAs, as well as cell spreading area ($P = 0.0014$, Supplementary Fig. 7), consistent with the previous reports[14]. Importantly, talin-1 depletion eliminates the linear relationship between force-tFAK (Fig. 2h, $P = 0.6061$) and force-pY397-FAK (Fig. 2i, $P = 0.2421$). Although we cannot conclude whether talin-2 compensates for talin-1 depletion, these results demonstrate that talin-1 depletion significantly reduces traction forces, FAK localization, and Y397 phosphorylation and, importantly, that talin-1 is required for linear force-FAK signaling coupling at individual FAs.

We next evaluated the role of FAK functional sites on force-FAK signaling coupling using FAK-null fibroblasts expressing eGFP-FAK wild-type (WT) and mutant constructs (Fig. 2j) on stiff (14 kPa) mPADs. Expression of WT and these mutant FAK proteins in FAK-null cells increases total traction force and spreading area compared to FAK-null controls (Supplementary Fig. 8). The FAK mutants localize to FAs to equivalent levels as WT FAK (Supplementary Fig. 9), except for the K454R kinase-dead mutant which has ~25% higher average levels than WT ($P = 0.0004$) and the E1015A talin-

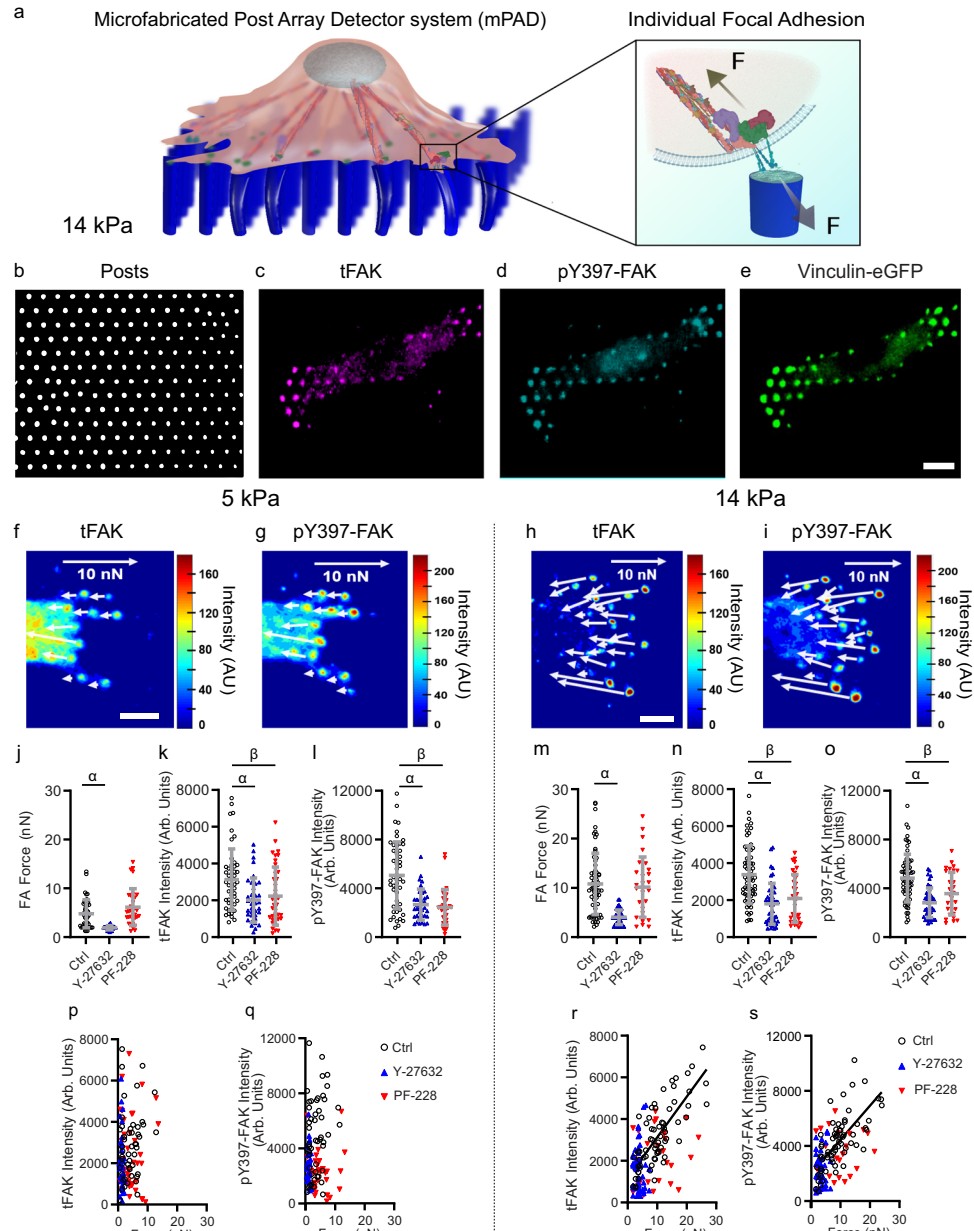

**Fig. 1 Force-FAK localization and force-pY397-FAK are linearly coupled at individual FAs. a** Schematic of individual FA traction force measurement for a cell using mPADs. Fluorescence images showing (**b**) mPAD posts, (**c**) tFAK, (**d**) pY397-FAK, and (**e**) eGFP-vinculin for cell on mPAD. **f–i** Heatmaps for (**f**, **h**) tFAK and (**g**, **i**) pY397-FAK staining at FAs on mPADs of 5 or 14 kPa. White arrows indicate force magnitudes and directions. Scale bar 5 μm. **j** Force (mean ± SD) at individual FAs on 5 kPa mPADs. Two-sided Kruskal-Wallis test $P < 0.0001$, $\alpha$: $P < 0.0001$, $n = 50, 36, 33$ FAs for each condition, respectively. **k** tFAK integrated intensity (mean ± SD) at FAs on 5 kPa mPADs. ANOVA $P < 0.0001$, $\alpha$: $P = 0.0032$, $\beta$: $P = 0.0195$, $n = 49, 37, 33$ FAs for each condition, respectively. **l** pY397-FAK integrated intensity (mean ± SD) at individual FAs on 5 kPa mPADs. Two-sided Kruskal-Wallis test $P < 0.0001$, $\alpha$: $P = 0.0002$, $\beta$: $P < 0.0001$, $n = 50, 36, 33$ FAs for each condition, respectively. **m** Force (mean ± SD) at individual FAs on 14 kPa mPADs. Two-sided Kruskal-Wallis test $P < 0.0001$, $\alpha$: $P < 0.0001$, $n = 66, 48, 28$ FAs for each condition, respectively. **n** tFAK integrated intensity (mean ± SD) at FAs on 14 kPa mPADs. ANOVA $P < 0.0001$, $\alpha$: $P < 0.0001$, $\beta$: $P = 0.0004$, $n = 66, 48, 33$ FAs for each condition, respectively. **o**, pY397-FAK integrated intensity (mean ± SD) at individual FAs on 14 kPa mPADs. Two-sided Kruskal-Wallis test $P < 0.0001$, $\alpha$: $P < 0.0001$, $\beta$: $P < 0.0146$, $n = 66, 46, 28$ FAs for each condition, respectively. **p** Force vs. tFAK intensity at individual FAs on 5 kPa. Linear regression: control $P = 0.1230$, $n = 50$ FAs; Y-27632 $P = 0.1920$, $n = 37$ FAs; PF-228 $P = 0.9869$, $n = 38$ FAs. **q**, Force vs. pY397-FAK intensity at individual FAs on 5 kPa. Linear regression: control $P = 0.1072$, $n = 50$ FAs; Y-27632 $P = 0.2937$, $n = 36$ FAs; PF-228 $P = 0.8874$, $n = 33$ FAs. **r** Force vs. tFAK intensity at individual FAs on 14 kPa. Linear regression: control $P < 0.0001$, $n = 66$ FAs, tFAK intensity $= 194.6 \times$ force $+ 1178$; Y-27632 $P = 0.8896$, $n = 48$ FAs; PF-228 $P = 0.2084$, $n = 28$ FAs. **s** Force vs. pY397-FAK intensity at individual FAs on 14 kPa. Linear regression: control $P < 0.0001$, $n = 66$ FAs, pY397-FAK intensity $= 219.7 \times$ force $+ 2027$; Y-27632 $P = 0.7986$, $n = 46$ FAs; PF-228 $P = 0.2260$, $n = 26$ FAs.

binding mutant ($P = 0.0005$). Cells expressing WT and E1015A exhibit equivalent average levels of pY397-FAK, whereas cells expressing the K454R and Y397F phosphorylation mutant display very low levels of pY397 staining (Supplementary Fig. 9). We

examined the relationship between traction force and tFAK, as well as traction force and pY397-FAK at individual FAs. Consistent with results for wild-type MEFs (Fig. 1r, s), WT FAK-expressing FAK-null cells exhibit force-tFAK and force-pY397-FAK linear

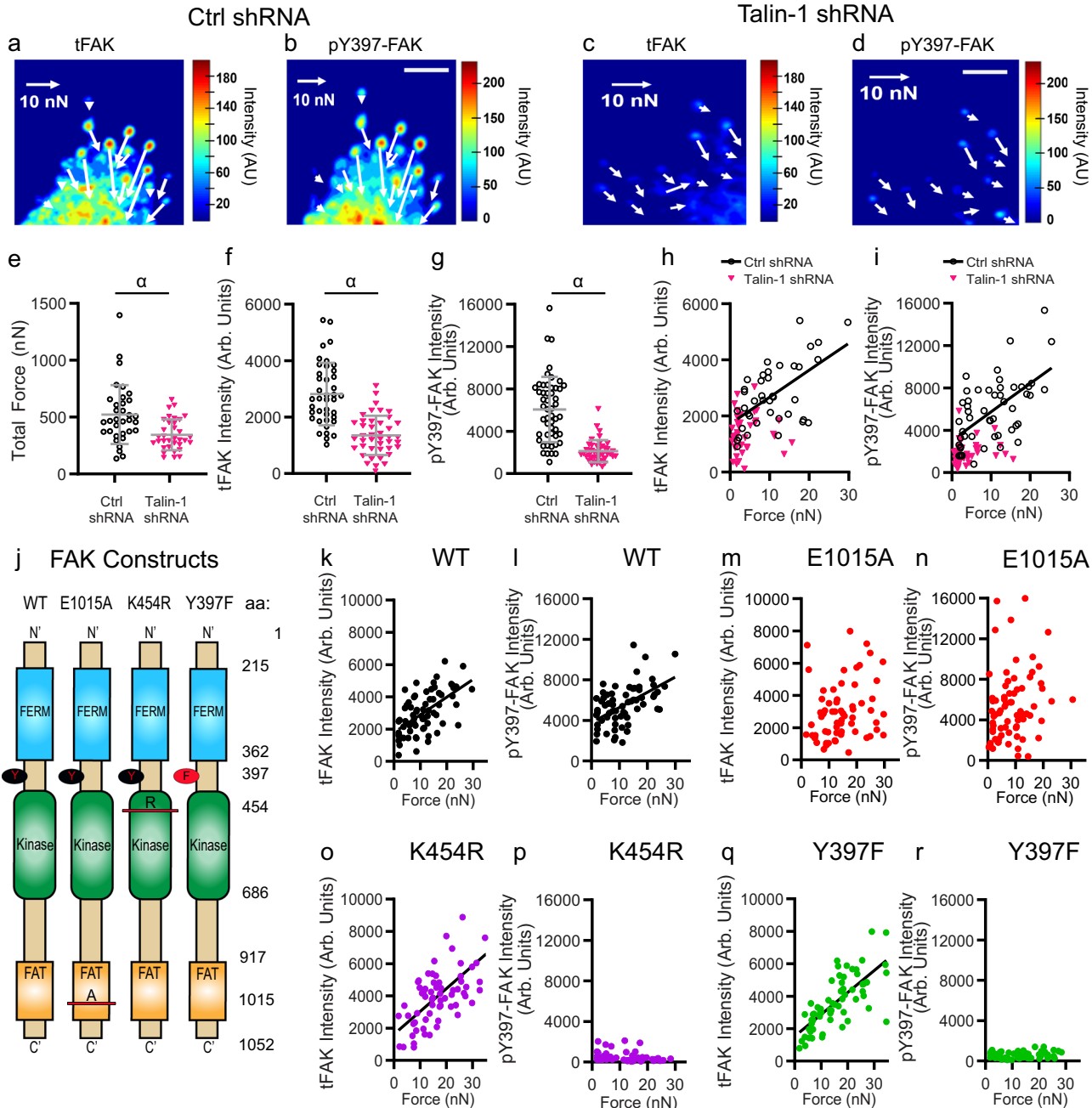

**Fig. 2 Talin regulates linear force-FAK signaling coupling at FAs. a–d** Heatmaps of tFAK and pY397-FAK intensities at FAs for cells treated with (**a**, **b**) control or (**c**, **d**) talin-1 shRNA. Scale bar 5 μm. **e** Total traction force (mean ± SD) for control shRNA-treated and talin-1 shRNA-treated cells. Two-sided Mann–Whitney test $\alpha$: $P = 0.0005$, $n = 34$ cells for each condition. **f** tFAK intensity (mean ± SD) at individual FAs. Two-sided Mann–Whitney test $\alpha$: $P < 0.0001$, $n = 40$ and 42 FAs for control and talin-1 shRNA, respectively. **g** pY397-FAK intensity (mean ± SD) at individual FAs. Two-sided Mann–Whitney test $\alpha$: $P < 0.0001$, $n = 51$ and 42 FAs for control and talin-1 shRNA, respectively. **h** Traction force vs. tFAK intensity at individual FAs. Linear regression: control shRNA $P < 0.0001$, $n = 40$ FAs, tFAK intensity $= 96.1 \times$ force $+ 1712$; talin-1 shRNA $P = 0.6061$, $n = 42$ FAs. **i** Force vs. pY397-FAK intensity at individual FAs. Linear regression: control shRNA $P < 0.0001$, $n = 51$ FAs, pY397-FAK intensity $= 268.3 \times$ force $+ 3056$; talin-1 shRNA $P = 0.2421$, $n = 42$ FAs. **j** Schematic of FAK constructs. **k** Force vs. tFAK intensity at individual FAs for WT FAK. Linear regression $P < 0.0001$, $n = 70$ FAs, tFAK intensity $= 110.2 \times$ force $+ 1779$. **l** Force vs. pY397-FAK intensity at individual FAs for WT FAK. Linear regression $P < 0.0001$, $n = 68$ FAs, pY397-FAK intensity $= 150.0 \times$ force $+ 3776$. **m** Force vs. tFAK intensity at individual FAs for E1015A FAK. Linear regression $P = 0.0607$, $n = 65$ FAs. **n** Force vs. pY397-FAK intensity at individual FAs for E1015A FAK. Linear regression $P = 0.0619$, $n = 78$ FAs. **o** Force vs. tFAK intensity at individual FAs for K454R FAK. Linear regression $P < 0.0001$, $n = 70$ FAs, tFAK intensity $= 142.1 \times$ force $+ 1607$. **p** Force vs. pY397-FAK intensity at individual FAs for K454R FAK. Linear regression $P = 0.0610$, $n = 48$ FAs. **q** Force vs. tFAK intensity at individual FAs for Y397F FAK. Linear regression $P < 0.0001$, $n = 65$ FAs, tFAK intensity $= 135.1 \times$ force $+ 1525$. **r** Force vs. pY397-FAK intensity at individual FAs for Y397F FAK. Linear regression $P = 0.1869$, $n = 60$ FAs.

relationships (Fig. 2k, l, linear regression force-tFAK $P < 0.0001$; force-pY397-FAK $P < 0.0001$). In contrast, FAK-null cells expressing the talin-binding mutant FAK E1015A do not show linear coupling between force and tFAK (Fig. 2m, $P = 0.0607$) and force and pY397-FAK (Fig. 2n, $P = 0.0619$) at individual FAs, even though the average levels of total traction force ($P = 0.0721$), tFAK localization ($P > 0.9999$), and pY397-FAK ($P = 0.2785$) are not different between WT and E1015A FAK-expressing cells (Supplementary Fig. 8, 9). This result demonstrates that FAK binding to talin is required for linear force-FAK signaling coupling at individual FAs and is fully consistent with the requirement of talin-1 for this mechanosensing response (Fig. 2h, i). FAK-null cells expressing the FAK K454R kinase-dead or the Y397F phosphorylation mutants show a linear relationship between force and tFAK (Fig. 2o, $P < 0.0001$; Fig. 2q, Y397F $P < 0.0001$), indicating that neither FAK catalytic activity nor phosphorylation of Y397 is necessary for force-tFAK linear coupling at individual FAs. This result differs from the results with the PF-228 inhibitor (Fig. 1n). Because PF-228 may have off-target effects[25], we conclude that FAK kinase activity is not required for tFAK localization to FAs based on the results for cells expressing the K454R mutant. However, the K454R kinase-dead mutant shows background levels of pY397-FAK (Fig. 2p), indicating that FAK kinase activity is necessary for phosphorylation of Y397 at FAs. As expected, cells expressing the Y397F phosphorylation mutant display background levels of pY397-FAK (Fig. 2r).

The requirements for substrate stiffness and actomyosin contractility in force-tFAK and force-pY397-FAK linear coupling at individual FAs suggest that a mechanical balance between traction forces and cytoskeletal tension controls force-FAK signaling coupling at FAs. We, therefore, analyzed the role of vinculin, an important force-transmitting protein that binds talin and actin at FAs[24], in force-FAK signaling coupling using vinculin-null fibroblasts expressing WT and mutant vinculin proteins (Fig. 3a). Expression of WT vinculin in vinculin-null cells increases the average traction force at FAs compared to control vinculin-null cells ($P = 0.0138$) and cells expressing a truncated vinculin head (VH) mutant that localizes to FAs but cannot bind actin[24] ($P < 0.0001$) or the talin binding-deficient A50I full-length mutant[26] ($P = 0.0058$, Fig. 3b). No differences in average levels of tFAK (Fig. 3c) at FAs were detected between WT vinculin-expressing cells and cells expressing the VH ($P > 0.9999$) or A50I mutant ($P = 0.2265$), although higher levels of tFAK were present at FAs for vinculin-null cells compared to cells expressing WT and VH vinculin. No differences were observed in pY397-FAK at FAs among vinculin cell lines (Fig. 3d, $P = 0.2276$). These data confirm that, whereas vinculin modulates traction force, this protein is not required for FAK localization or phosphorylation of Y397-FAK at FAs[27]. Nevertheless, vinculin-null cells expressing WT vinculin exhibit force-tFAK and force-pY397-FAK linear coupling (Fig. 3e, f, linear regression force-tFAK $P < 0.0001$, force-pY397-FAK $P < 0.0001$). In contrast, vinculin-null controls (Fig. 3g, h, force-tFAK $P = 0.7124$; force-pY397-FAK $P = 0.8753$) and cells expressing VH (Fig. 3i, j, force-tFAK $P = 0.4426$; force-pY397-FAK $P = 0.5564$) or A50I (Fig. 3k, l, force-tFAK $P = 0.8549$; force-pY397-FAK $P = 0.1973$) mutants do not exhibit linear relationships for force-tFAK and force-pY397-FAK. These results demonstrate that a full-length vinculin molecule that binds talin and actin is required for linear force-FAK signaling coupling at FAs.

**FAK phosphorylation increases linearly with applied external force**. The mPADs system provides a robust and experimentally convenient platform to examine the relationship between traction force and FAK signaling at individual FAs on deformable substrates with defined mechanical properties. Nevertheless, the linear relationships for force-tFAK and force-pY397-FAK on stiff substrates are generated from correlative data based on mPAD

post deflections and immunostaining. To gain further insights on force-FAK signaling coupling at FAs, we used fibronectin-coated magnetic beads (4.5 μm diameter) to apply external forces to integrin-based FAs and isolate loaded adhesive complexes[28,29] (Fig. 4a). After incubating adherent cells with fibronectin-coated magnetic beads for 40 min, a permanent magnet was placed at a defined height over the cells for 10 min to apply external tensile forces to FAs associated with the magnetic beads. After cell lysis, magnetic beads and associated adhesive complexes were isolated by magnetic separation, dissociated, and analyzed for tFAK and pY397-FAK levels by Western blot. Cells held in suspension for 1 h prior to Western blot analyses served as negative controls with background levels of pY397-FAK. We first analyzed FAK-null fibroblasts expressing FAK constructs. For cells expressing WT FAK, incubation of beads with no applied magnetic force results in a significant increase in pY397-FAK/tFAK levels compared to suspension cells (Fig. 4b, $P = 0.0013$). Importantly, the levels of pY397-FAK/tFAK increase linearly with applied magnetic force (Fig. 4b, linear regression $P = 0.0489$). As expected, FAK-null cells have background levels of pY397FAK/FAK (Fig. 4c). For FAK-null cells expressing the E1015A talin-binding FAK mutant, incubation of beads with no applied magnetic force results in a significant increase in pY397-FAK/tFAK levels compared to suspension cells (Fig. 4d, $P = 0.0158$). However, there is no relationship between applied magnetic force and pY397-FAK/tFAK levels (Fig. 4d, $P = 0.0578$).

We next examined the relationship between applied force and Y397-FAK phosphorylation in vinculin-null fibroblasts expressing vinculin constructs using the magnetic bead platform. For vinculin-null cells expressing WT vinculin, pY397-FAK/tFAK levels increased linearly with applied force (Fig. 4e, linear regression $P = 0.0288$). In contrast, vinculin-null cells and vinculin-null cells expressing the A50I vinculin mutants exhibit reduced pY397-FAK/tFAK levels, and applied magnetic force has no effect on pY397-FAK/tFAK levels (Fig. 4f, $P = 0.4776$; Fig. 4g, $P = 0.5468$). Taken together, these results with the magnetic bead platform demonstrate that Y397-FAK phosphorylation levels increase linearly with applied force, and talin-FAK binding and vinculin are required for this linear coupling. Importantly, these results are in excellent agreement with the data obtained in the mPADs system. A limitation of the mPADs system is that the FA area is limited to the size of the micropost (1.8 μm diameter). The magnetic bead (4.5 μm diameter sphere) provides a ~12-fold increase in available area for FAs, and the high concordance in results between these two platforms suggests that the linear force-FAK signaling coupling is not limited to the small available area of mPADs. In addition, we estimated the magnetic puller stiffness by dividing the applied bead forces by the expected range of displacements (0.3–1 μm)[29], yielding spring constant values in the order of 0.03 nN/μm. This estimate is approximately 2 orders of magnitude lower than the spring constant for the mPADs (~7 nN/μm), suggesting that the observed FAK responses in the magnetic bead experiments are attributable to force and not stiffness.

**Force-FAK signaling coupling coordinates microtissue repair**. To investigate the biological significance of force-FAK signaling coupling, we employed an in vitro biomimetic wound healing model using microfabricated tissue constructs[30] (Fig. 5a). In this platform, a 3D collagen gel seeded with cells is suspended between flexible cantilevers within a microfabricated mold, and encapsulated cells generate traction forces to contract the collagen gel into a dense fibrocellular microtissue attached to the deformable posts. A microsurgically-induced defect (i.e., wound) in the center of the microtissue is rapidly closed by coordinated cell forces and

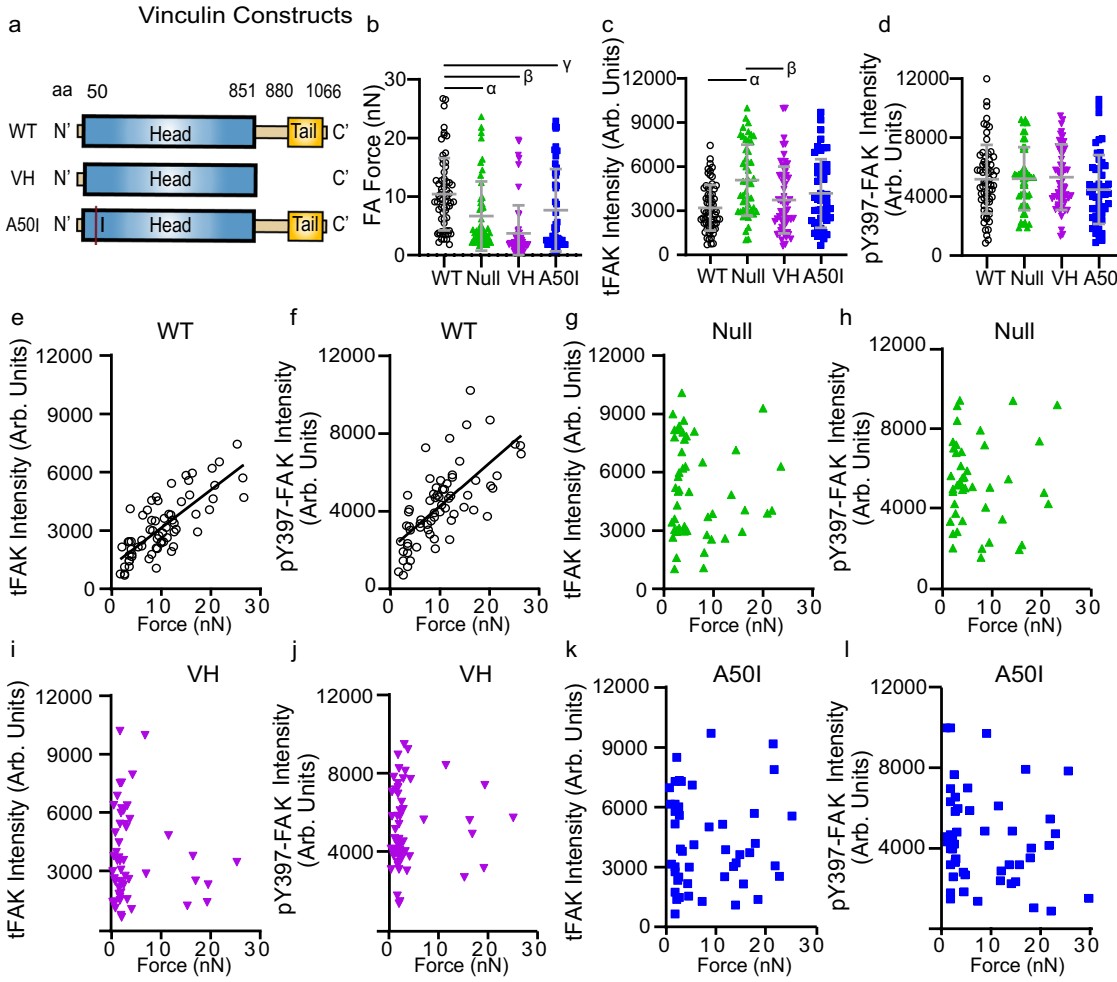

**Fig. 3 Vinculin mediates linear force-FAK signaling coupling at FAs. a** Vinculin constructs. **b** Traction force (mean ± SD) at individual FAs. Two-sided Kruskal-Wallis test $P < 0.0001$, $\alpha$: $P = 0.0138$, $\beta$: $P < 0.0001$, $\gamma$: $P = 0.0058$, $n = 66, 49, 57, 48$ FAs for each condition, respectively. **c** tFAK intensity (mean ± SD) at individual FAs. Two-sided Kruskal-Wallis test $P = 0.0003$, $\alpha$: $P = 0.0002$, $\beta$: $P < 0.0084$, $n = 66, 49, 57, 49$ FAs for each condition, respectively. **d** pY397-FAK intensity (mean ± SD) at individual FAs. ANOVA $P=0.2276$, $n = 66, 46, 57, 49$ FAs for each condition, respectively. **e** Force vs. tFAK intensity at individual FAs for WT vinculin. Linear regression $P < 0.0001$, $n = 66$ FAs, tFAK intensity $= 194.6 \times$ force $+ 1178$. **f** Force vs. pY397-FAK intensity at individual FAs for WT vinculin. Linear regression $P < 0.0001$, $n = 66$ FAs, pY397-FAK intensity $= 219.7 \times$ force $+ 2027$. **g** Force vs. tFAK intensity at individual FAs for vinculin-null cells. Linear regression $P = 0.7124$, $n = 49$ FAs. **h** Force vs. pY397-FAK intensity at individual FAs for vinculin-null cells. Linear regression $P = 0.8753$, $n = 48$ FAs. **i** Force vs. tFAK intensity at individual FAs for VH vinculin. Linear regression $P = 0.4426$, $n = 57$ FAs. **j** Force vs. pY397-FAK intensity at individual FAs for VH vinculin. Linear regression $P = 0.5564$, $n = 57$ FAs. **k** Force vs. tFAK intensity at individual FAs for A50I vinculin. Linear regression $P = 0.8549$, $n = 49$ FAs. **l** Force vs. pY397-FAK intensity at individual FAs for A50I vinculin. Linear regression $P = 0.1973$, $n = 49$ FAs.

migration, providing an in vitro model of 3D fibrous tissue repair. We examined the tissue repair response for FAK-null fibroblasts expressing WT FAK or the E1015A talin-binding FAK mutant or control FAK-null cells (Supplementary movies 1–3). For microtissues containing WT FAK-expressing cells, the wound area rapidly increases following injury due to pre-stress arising from cell force-driven tissue compaction, but the wound area then decreases monotonically as the tissue repairs until it is fully closed (Fig. 5b). The profile for wound area, normalized to the initial wound area, overtime is accurately described by a log-normal curve (Fig. 5b), and this curve fit was used to estimate the amplitude and wound closure rate parameters to describe the tissue healing response (Supplementary Fig. 10). In stark contrast to WT FAK-expressing cells, wounds for microtissues containing FAK-null cells maintain constant area over time and do not close (Fig. 5b), demonstrating that FAK is required for microtissue repair. Microtissues containing cells expressing the E1015A FAK mutant exhibit delayed wound closure and several microtissues did not fully repair wounds

(Fig. 5b). Analysis of curve-fit parameters confirms these observations. Microtissues containing WT FAK-expressing cells exhibit higher amplitude, reflecting higher pre-stress prior to the injury, compared to microtissues seeded with E1015A FAK-expressing cells ($P = 0.0440$) and FAK-null cells (Fig. 5c, $P = 0.0176$). Furthermore, microtissues containing WT FAK-expressing cells display faster closure rates compared to microtissues seeded with E1015A FAK-expressing cells ($P = 0.0213$) and FAK-null cells (Fig. 5d, $P = 0.0088$). The wound closure rate for microtissues seeded with E1015A FAK-expressing cells is higher than the rate for tissues with FAK-null cells ($P = 0.0289$). At 24 h post-wounding, microtissues seeded with WT FAK-expressing cells generate higher contractile forces compared to microtissues containing E1015A FAK-expressing cells ($P = 0.0266$) and FAK-null cells (Fig. 5e, $P = 0.0003$). In addition, microtissues containing WT FAK-expressing or E1015A FAK-expressing cells exhibit higher width contraction along the centerline compared to microtissues seeded with FAK-null cells (Fig. 5f, $P = 0.0015$, $P = 0.0101$). Microtissues containing

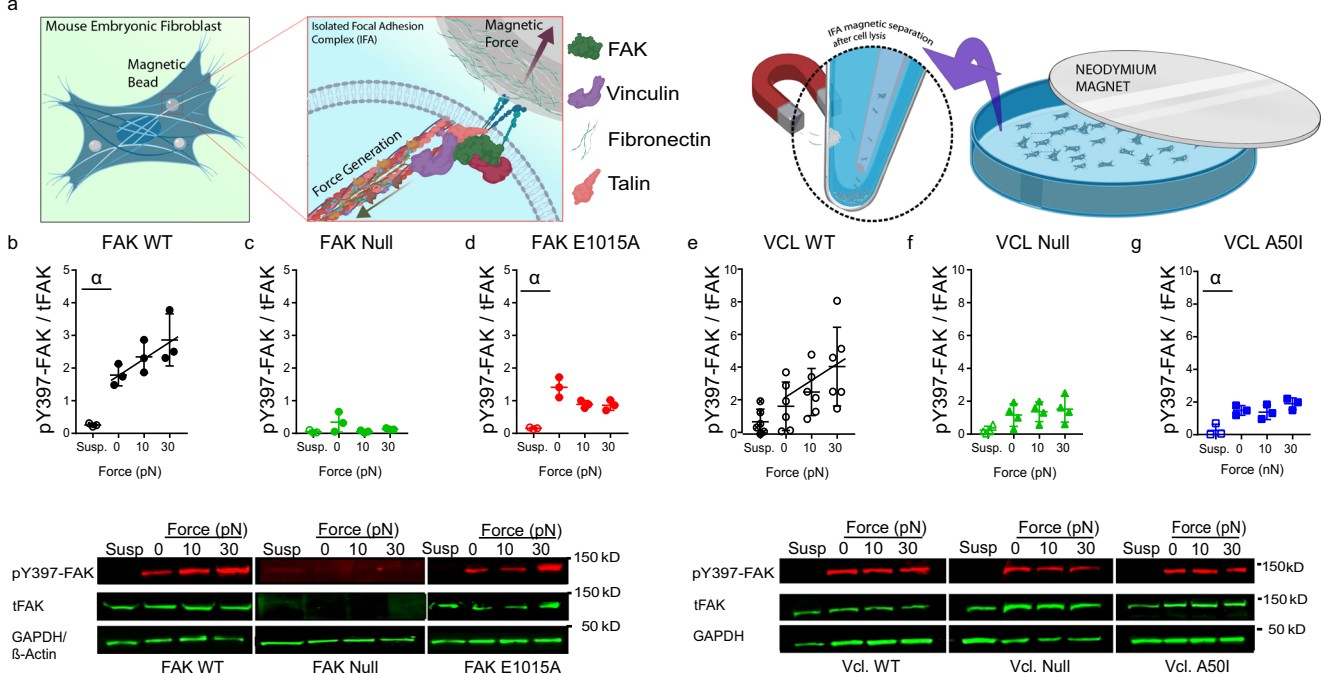

**Fig. 4 FAK phosphorylation increases linearly with applied force. a** Schematic of application of external force to integrin-based FAs and isolation of FAs using magnetic beads. **b-d,** pY397-FAK/tFAK levels for FAK-null cells expressing FAK constructs (top) and Western blots (below). **b** For WT FAK-expressing cells, pY397-FAK/tFAK levels (mean ± SD) increase linearly with applied force. Linear regression ($P = 0.0489$, pY397-FAK/FAK = 0.0345 × force + 1.870). t-test $\alpha$: $P = 0.0013$, $n = 3$ independent experiments. **c** For FAK-null cells, background levels of pY397-FAK/tFAK (mean ± SD) are detected ($n = 3$ independent experiments). **d** For cells expressing FAK E1015A, pY397-FAK/tFAK levels (mean ± SD) are independent of external force. Linear regression ($P = 0.0578$). t-test $\alpha$: $P = 0.0158$, $n = 3$ independent experiments. **e–g** pY397-FAK/tFAK levels for vinculin-null cells expressing vinculin constructs (top) and Western blots (below). **e** For WT vinculin-expressing cells, pY397-FAK/tFAK levels (mean ± SD) increase linearly with applied force. Linear regression $P = 0.0288$, pY397-FAK/FAK = 0.0801 × force + 1.721, $n = 6$ independent experiments. **f** pY397-FAK/tFAK levels (mean ± SD) in vinculin-null cells are independent of external force. Linear regression $P = 0.4776$, $n = 4$ independent experiments. **g** For cells expressing FAK E1015A, pY397-FAK/tFAK levels (mean ± SD) are independent of external force. Linear regression $P = 0.5468$, $n = 3$ independent experiments.

WT FAK-expressing cells display higher levels of pY397-FAK intensity compared to microtissues seeded with E1015A FAK-expressing cells ($P = 0.0232$) and FAK-null cells (Fig. 5g, $P = 0.0020$). These results demonstrate that FAK expression and FAK-talin binding are required for tissue-scale force generation and proper microtissue repair. Consistent with this conclusion, microtissues containing talin-1-depleted cells exhibit reduced amplitude values, indicating reduced tissue pre-stress, and impaired wound closure rates compared to microtissues seeded with control cells (Supplementary Fig. 11).

Chen and colleagues demonstrated that microtissue wound closure requires coordinated cell migration[30]. Using automated cell tracking and image analysis, we measured individual cell trajectories during microtissue wound healing (Supplementary Fig. 12). Mean square displacement (MSD), cell speed, and straightness index (directionality index) were calculated from individual cell trajectories using MotilityLab (Supplementary Fig. 12). This analysis reveals lower migration speed and straightness index for WT FAK-expressing cells compared to FAK-null cells ($P < 0.0001$, $P < 0.0001$) and cells expressing the E1015A FAK mutant ($P = 0.0003$, $P = 0.0015$). Speed and straightness parameters are also different between E1015A FAK-expressing and FAK-null cells ($P = 0.0025$, $P = 0.0051$). Furthermore, the MSD data were curve-fitted to the Persistent Random Walk model to extract values for diffusivity and directional persistence. WT FAK-expressing cells display reduced persistence and diffusivity values compared to E1015A FAK-expressing ($P = 0.0202$, $P = 0.0284$) and FAK-null ($P = 0.0009$, $P = 0.0006$) cells. Taken together, these migration parameters show

that WT FAK-expressing cells in microtissues exhibit slower and more directed cell motions compared to E1015A FAK-expressing and FAK-null fibroblasts in microtissues. These results demonstrate that FAK-talin binding is required for coordinated and directed cell migration for microtissue wound closure.

**Model of talin-FAK binding under force.** We developed a simple kinetic model to simulate talin-FAK binding interactions under force to gain further insights into force-FAK signaling coupling (Fig. 6a). In this model, talin at FAs undergoes a reversible structural change from an unstretched to a stretched conformation under force. The effect of tension across talin is incorporated into the kinetic rate-controlling talin conversion into the stretched state using the Bell model[31] and in agreement with experimental measurements of talin stretching under force[13]. The stretched talin reversibly binds FAK to form a FAK-talin complex, and then undergoes reversible phosphorylation. The law of mass action and conservation laws were applied to derive a system of coupled differential equations describing the time-dependent changes for the number of stretched talin molecules, FAK-talin complexes, and phosphorylated FAK-talin complexes. Solutions at steady state were obtained numerically as a function of the force applied to talin over a range of forces consistent with measurements in live cells (7–11 pN)[22], and parametric analyses were performed for relevant conditions. We first explored the influence of the coupling factor $\alpha$ that relates the force applied to talin to the forward rate-controlling conversion into the talin stretched state ($k_1$). For $\alpha = 0.05$, a value in agreement with experimental measurements of talin unfolding

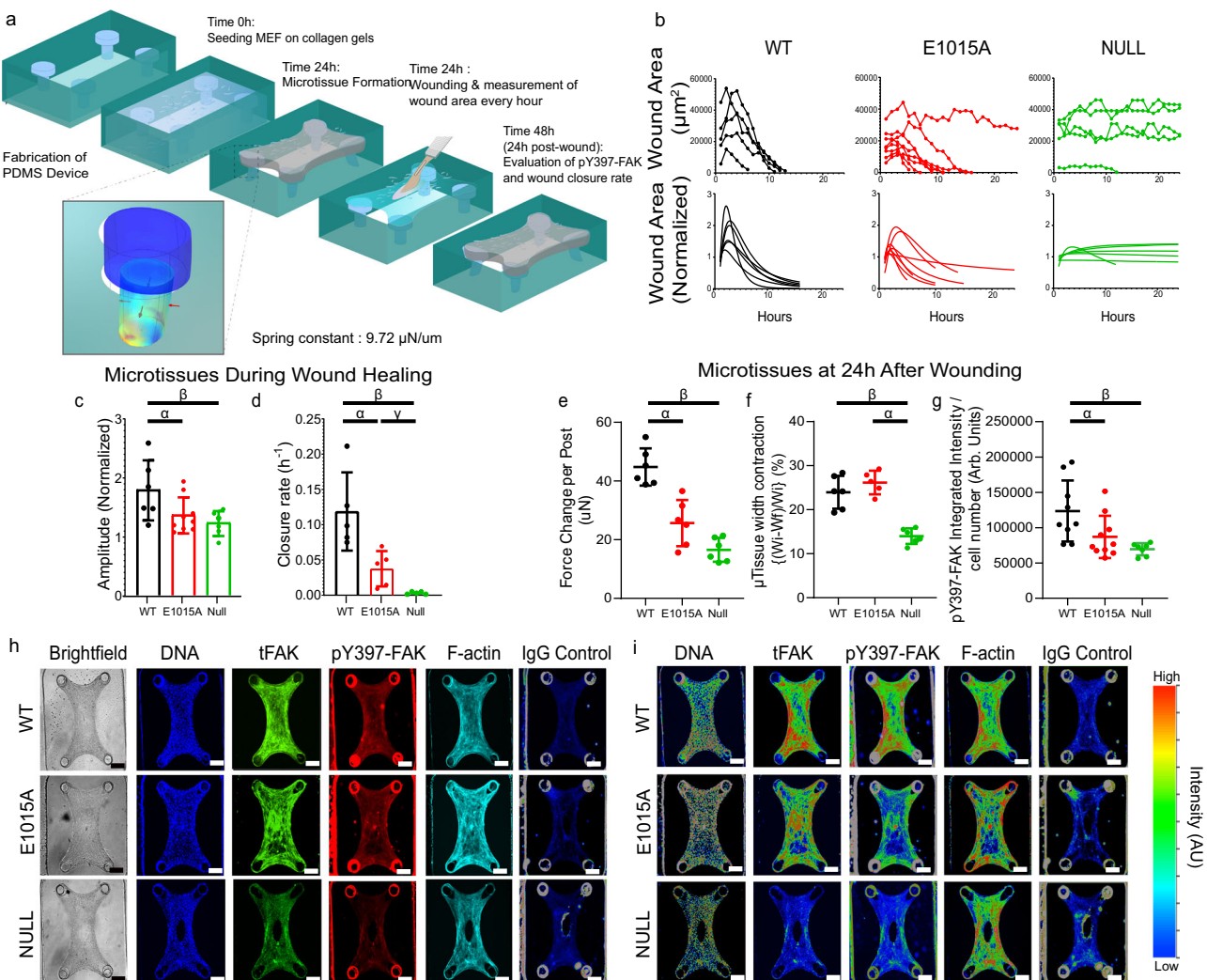

**Fig. 5 FAK-talin binding regulates wound closure of 3D microtissues. a** Schematic of microtissue assembly and wound closure assay. Collagen gels containing FAK-null cells expressing FAK constructs compact and form microtissue. At 24 h post-seeding, the wound is created using a microdissection knife, and wound closure is monitored over time. **b** Wound area (top) and log-normal curve fit for wound area normalized to the initial area (bottom) as a function of time for individual microtissues. **c** Amplitude (mean ± SD) of normalized wound area profile. Kruskal-Wallis test $P = 0.0351$, $\alpha$: $P = 0.0440$, $\beta$: $P = 0.0176$, $n = 6$ microtissues for WT and null and 9 microtissues for E1015A. **d** Closure rate (mean ± SD) of normalized wound area profile. ANOVA with Welch's correction $P = 0.0064$, $\alpha$: $P = 0.0213$, $\beta$: $P = 0.0088$, $\gamma$: $P = 0.0289$, $n = 5$ microtissues for each condition. **e**, Net force difference (mean ± SD) between pre-wounding and 24 h post-wounding. Kruskal-Wallis test $P < 0.0001$, $\alpha$: $P = 0.0266$, $\beta$: $P = 0.0003$, $n = 6$ microtissues for each condition. **f** Microtissue width contraction (mean ± SD). Kruskal-Wallis test $P = 0.0002$, $\alpha$: $P = 0.0015$, $\beta$: $P = 0.0101$, $n = 6$ microtissues for each condition. **g** pY397-FAK integrated intensity (mean ± SD) at 24 h post-wounding. Kruskal-Wallis test $P = 0.0056$, $\alpha$: $P = 0.0232$, $\beta$: $P = 0.0020$, $n = 9, 10$, and 7 microtissues for WT, E1015A, and null, respectively. **h** Brightfield and fluorescence (DNA, tFAK, pY397-FAK, F-Actin, IgG control) images for microtissues at 24 h post-wounding. **i** Intensity heatmaps (DNA, tFAK, pY397-FAK, F-Actin, IgG control) for microtissues. The images in **h**–**e** are representative of images from $n = 9, 10$, and 7 microtissues for respective groups. Scale bar 200 μm.

under force[13], the number of stretched talin molecules, FAK-talin complexes, and phosphorylated FAK-talin complexes increase linearly with force applied to talin (Fig. 6b–d). The predicted linear increases in FAK-talin complexes and phosphorylated FAK-talin complexes with force are in good agreement with our experimental observations for the linearity of force-FAK signaling coupling at individual FAs on stiff substrates (Fig. 1r, s). For smaller values of $\alpha$, the number of stretched talin molecules, FAK-talin complexes, and phosphorylated FAK-talin complexes is independent of the force applied to talin. These simulations are in line with our experimental results showing that inhibition of actomyosin contractility (Fig. 1r, s) or adhesion to soft substrates (Fig. 1p, q) eliminates the linear relationship for force-FAK signaling at FAs. Smaller values of $\alpha$ correspond to kinetic rates of switching between unstretched and stretched talin conformations that are insensitive to force, and

therefore reflect a talin molecule that does not respond to force. Consistent with this explanation, Grashoff and Schwartz independently reported that inhibition of contractility or adhesion to soft substrates reduces tension across talin[22,32].

Our experimental findings demonstrate that the talin-binding site in FAK is required for linear force-FAK signaling coupling (Fig. 2m, n), mechanosensing (Fig. 4d), and microtissue repair (Fig. 5b–e). Mutation of this site can be modeled as a reduction in $k_3$, the forward binding rate for the FAK-talin complex. Computer simulations show that decreases in the value of $k_3$ considerably reduce the number of FAK-talin complexes and phosphorylated FAK-talin complexes (Fig. 6f, g), and a 100-fold reduction in $k_3$ eliminates the dependence of the number of phosphorylated FAK-talin complexes on the force applied to talin. Conversely, the mutation that impacts FAK-talin binding

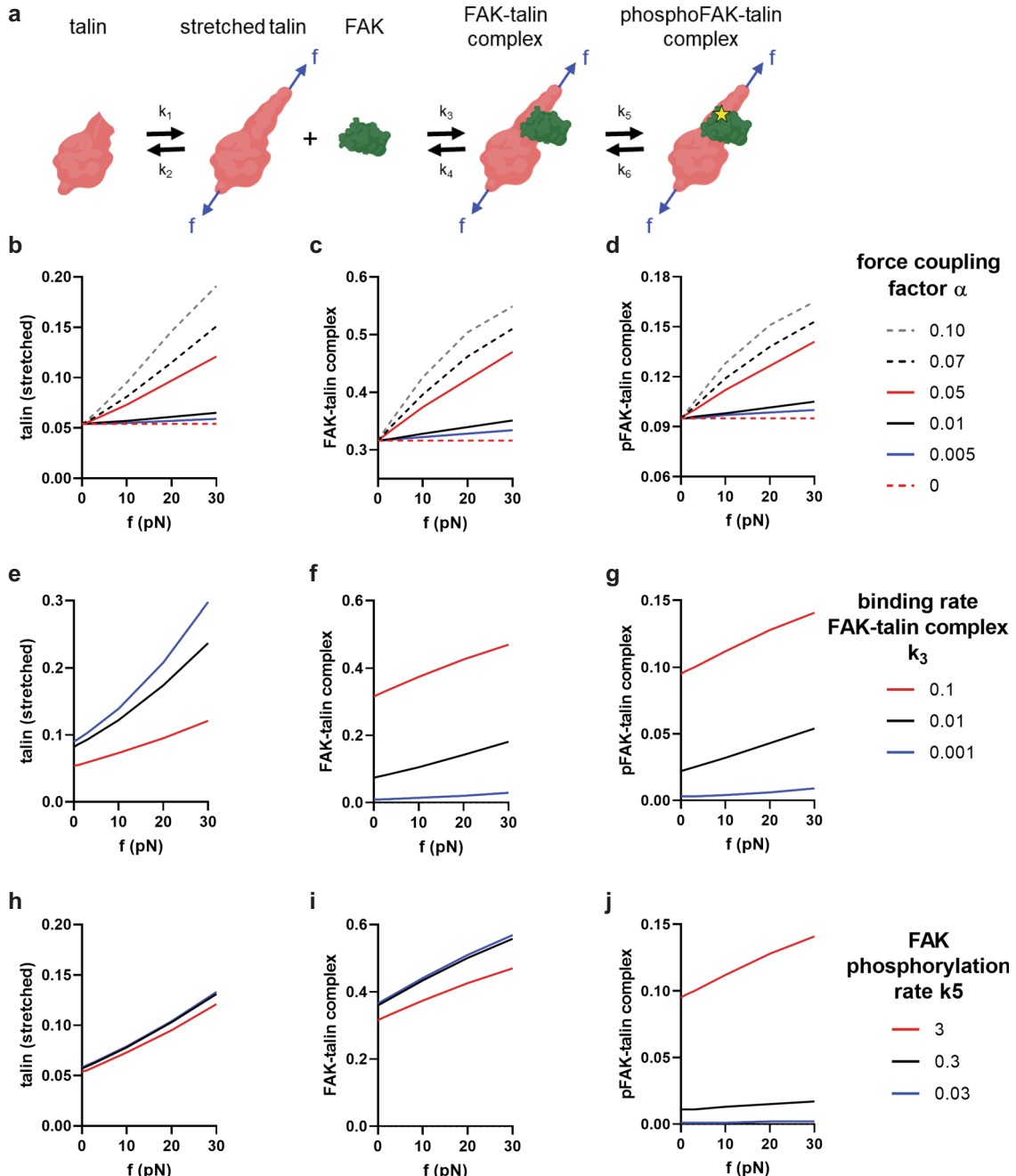

**Fig. 6 Kinetic model of talin-FAK interactions under force. a** Schematic of FAK-talin interactions showing unstretched ($T$) and stretched talin ($T_{st}$), free FAK ($F$), FAK-talin complex ($FT_{st}$), and phosphorylated FAK-talin complex ($pFT_{st}$). Governing equations and parameters used in the kinetic model are listed in Supplementary Materials. **b-d** Solutions for a relative number of (**b**) stretched talin, (**c**) FAK-talin complexes, and (**d**) phosphorylated FAK-talin complexes as a function of applied force (f) for different values of the force coupling factor ($\alpha$). **e-g** Solutions for a relative number of (**e**) stretched talin, (**f**) FAK-talin complexes, and (**g**) phosphorylated FAK-talin complexes as a function of applied force (f) for different values of the forward binding rate constant for FAK-talin complex ($k_3$). **h-j** Solutions for a relative number of (**h**) stretched talin, (**i**) FAK-talin complexes, and (**j**) phosphorylated FAK-talin complexes as a function of applied force (f) for different values of the FAK phosphorylation rate constant ($k_5$).

can be modeled by increases in the reverse binding rate for the FAK-talin complex ($k_4$). As expected, increasing $k_4$ has equivalent effects on the number of FAK-talin complexes and phosphorylated FAK-talin complexes as decreasing $k_3$. We conducted additional simulations to examine the effect of the FAK phosphorylation rate $k_5$. As expected, decreases in $k_5$ drastically reduce the number of phosphorylated FAK-talin complexes and eliminate its dependence on force, but decreases in $k_5$ have relatively modest effects on the linear relationship between

applied force and the number of stretched talin molecules and FAK-talin complexes (Fig. 6h–j). These simulations mirror the experimental results for cells expressing the FAK K454R kinase-dead mutant showing a linear relationship between traction force and tFAK and background levels of pY397-FAK (Fig. 2o, p).

The kinetic binding model does not explicitly consider vinculin's role in linear force-FAK signaling coupling. Based on the requirement of the talin-binding site in vinculin for linear force-FAK signaling coupling (Fig. 3k, l), we posit that vinculin

binds to and stabilizes the stretched talin conformation while transmitting force to actin. As such, the contributions of vinculin can be lumped within the kinetic rates of switching between unstretched and stretched talin conformations. Several lines of evidence support this idea. First, tension across talin exposes cryptic binding sites for vinculin binding[13]. Second, tension across talin at FAs is decreased in the absence of vinculin but re-expression of full-length vinculin restores mechanical loading of talin[22]. Interestingly, full-length vinculin that binds talin and actin is needed for both mechanical loading of talin[22] and linear force-FAK signaling coupling (Fig. 3e, f).

## Discussion

Prior work has shown that treatment with inhibitors of contractility significantly reduces Y397-FAK phosphorylation, indicating that traction force is required for FAK phosphorylation. These studies examined average levels of FAK phosphorylation and did not examine the functional/mathematical relationship between traction force levels and FAK phosphorylation. By analyzing individual FAs in cells adhering to deformable micropillar arrays, we demonstrate that traction force and FAK localization and traction force and Y397-FAK phosphorylation are linearly coupled at individual FAs on stiff, but not soft, substrates. The linear coupling between traction force and FAK localization and traction force and Y397-FAK phosphorylation requires actomyosin contractility, talin-FAK binding, and a full-length vinculin molecule that binds talin and actin. These linear relationships are not apparent when evaluating average population-based levels of force, FAK localization, and Y397-FAK phosphorylation but are only revealed by analyzing these metrics at individual FAs. Importantly, disruption of talin binding to FAK using the E1015A FAK mutant eliminates force-FAK localization and force-Y397-FAK phosphorylation even though this mutation does not alter average levels of traction force, FAK localization, and Y397-FAK phosphorylation at FAs. To our knowledge, traction force-FAK signaling linear coupling at individual FAs and the requirement of talin-FAK binding in these linear mechanosignaling relationships have not been previously reported. We further demonstrate that force-FAK coupling regulates two important cellular responses: (i) signaling in response to applied external forces using magnetic beads, and (ii) coordinated cell migration and tissue-scale forces driving microtissue repair using an in vitro 3D biomimetic wound healing model of fibrous tissue. Consistent with the results on micropost arrays, mutation of the talin-binding site in FAK significantly disrupts these two cellular processes.

We demonstrate a critical role for FAK in the microtissue repair process as significant defects in microtissue force, contraction, wound closure rate, and coordinated cell migration are evident for microtissues containing E1015A FAK-expressing and FAK-null cells compared to microtissues seeded with wild-type FAK-expressing cells. Consistent with the force-FAK linear coupling data generated from the micropost arrays, this analysis reveals an important role for FAK-talin binding in this integrated tissue repair response. We observed significant differences between microtissues containing wild-type FAK-expressing and E1015 FAK-expressing cells for microtissue force prior to wounding (as ascertained by the amplitude of the wound area profile) and during healing, wound closure rate, pY397-FAK integrated intensity in microtissue, and the cell migration parameters speed, straightness index, persistence time, and diffusivity. In addition, we found significant differences in wound healing rate, migration speed, and straightness (but not force, persistence, and diffusivity) between microtissues containing E1015A-expressing and FAK-null cells, indicating important contributions from FAK apart

from talin-binding interactions. These results demonstrate that FAK-talin binding is required for coordinated and directed cell migration for microtissue wound closure. Using a conventional wound scratch assay, Lawson et al. reported reduced wound closure for FAK E1015A-expressing cells compared to wild-type FAK-expressing cells, and no differences between FAK E1015A-expressing cells and FAK-null controls[33]. The packed cell-cell configuration and planar characteristics of these scratch models better recapitulate wound healing of epithelial sheets that heal by lamellipodial protrusions and purse-string contraction. In contrast, fibroblasts migrating in 3D collagen matrices do not display a lamellipodium, and migration is driven by mechanosensitive proteins that rely on actomyosin contractility[34]. In this study, we used a biomimetic 3D in vitro model of fibrous tissue repair with cells dispersed within a collagen construct in which wound healing involves coordinated tissue-scale forces, matrix assembly, and cell migration to restore 3D tissue architecture. In contrast to the reduced cell migration reported by Lawson et al., we observed increased cell speed, straightness index, persistence time, and diffusivity for FAK E1015A-expressing cells compared to wild-type FAK-expressing cells. Furthermore, we observed higher contractile forces in microtissues containing wild-type FAK-expressing cells compared to microtissues seeded with FAK E1015A-expressing cells. We also show differences in force, wound closure rate, and cell migration between microtissues containing FAK E1015A-expressing compared to microtissues seeded with FAK-null cells. These results highlight that wound closure in the 3D fibrous tissue repair model involves different mechanisms than the planar wound scratch model.

We present a simple kinetic binding model of talin-FAK binding interactions under a force that recapitulates the experimental observations. Our model proposes force-dependent structural changes in talin that regulate its interaction with FAK. Although talin at FAs is stretched by forces[22,32] and stretching modulates vinculin binding[13], to our knowledge, there is no direct experimental evidence showing that FAK binding to talin requires a mechanical stretch. In addition, our model does not consider the direct effects of force on FAK. Lietha et al. reported that force induces structural changes in FAK to pull the kinase domain away from the FERM domain[35]. While we cannot rule out a direct role of force on FAK activation, we speculate that the signaling events examined here occur after FAK adopts an open conformation on the cell membrane because the E1015A FAK mutant is still able to localize to FAs and become phosphorylated at Y397. We formulated the model with force as the input constant for direct correspondence to the metric (traction force) presented in the experimental results of the study. However, the traction force is a dynamic variable emerging as an output from the motor-clutch system[36], which provides a mechanistic explanation for the increasing force with increasing stiffness[37]. Because talin unfolding-mediated adhesion reinforcement via vinculin can be layered onto the base motor-clutch model to predict mechanotransduction phenomena[14], the current model could be integrated with this earlier framework to incorporate the effects of matrix stiffness.

These experimental and modeling analyses provide insights into how talin, FAK, and vinculin convert forces into early signaling events regulating mechanotransduction. This conceptual framework is relevant to adhesive force-signaling coupling at migratory cell fronts, force-regulated morphogenesis, and stem cell lineage commitment in response to matrix stiffness. Furthermore, this fundamental understanding of mechanosignaling can ultimately be exploited to design cell-biomaterial interactions. Exemplary applications include the design of biomaterial tools for basic cell biology studies, the development of culture supports for therapeutic

cell manufacturing, and the engineering of synthetic stem cell niches for regenerative medicine.

## Methods

**Cells and reagents**. eGFP-WT-vinculin, eGFP-VH-vinculin, eGFP-A50I-vinculin, and vinculin-null MEFs were generated by transducing vinculin-null MEFs with retroviral constructs for eGFP-vinculin variants and sorted by flow cytometry for eGFP expression, and cells have equivalent expression levels as vinculin[+/+] murine embryonic fibroblasts[38]. Cells were maintained in DMEM containing 10% FBS, 1% sodium pyruvate, and 1% penicillin-streptomycin. WT GFP-FAK, E1015A GFP-FAK, K454R GFP-FAK (kinase-dead), Y397F GFP-FAK, and FAK-null MEFs were derived by transducing FAK[−/−] MEFs (isolated from FAK[(loxP/loxP)] mice crossed with p53-null mice and transduced with AdCre) with lentiviral constructs for FAK variants, sorted by flow cytometry for GFP, and maintained as the pooled population under puromycin selection[21,33]. Talin-1 was depleted using talin-1 shRNA lentiviral particles (Santa Cruz Biotechnology) in DMEM containing 6 µg/mL polybrene. Control cells were treated with scrambled shRNA lentiviral particles (Santa Cruz Biotechnology). After the addition of lentiviral particles, cells were centrifuged at $1200 \times g$ for 30 min in a swinging bucket rotor and then incubated for 24 h at 37 °C and 5% $CO_2$. Cell culture media was aspirated and replaced with fresh media. Transduced cells were then selected using 4 µg/mL puromycin for 3-4 days, expanded, and either used for experimentation or cryopreserved. Depletion of talin-1 expression was confirmed by Western blotting. For studies with the eGFP-A50I-vinculin construct[24], $2 \times 10^6$ vinculin-null cells were transfected with 1.5 µg of the A50I construct via the Amaxa nucleofection kit (Lonza, Kit 2, Program T-020). Human mesenchymal stem cells were acquired from the NIH Resource Center at Texas A&M University. Cells were obtained under Texas A&M University IRB-approved protocols with informed consent from all human participants following relevant ethical regulations and provided as de-identified frozen samples.

**mPADs analyses**. Microfabricated post array detectors (mPADs) device silicon masters were fabricated using PDMS replica molding[38,39]. To make microfabricated post array templates, 1:10 PDMS prepolymer was cast on top of silanized mPADs device silicon masters, cured at 110 °C for 30 min, peeled off gently, treated with oxygen plasma (Plasma-Preen; Terra Universal), and silanized overnight with (tridecafluoro-1,1,2,2,-tetrahydrooctyl)-1-trichlorosilane (Sigma–Aldrich) vapor under vacuum. To make the final mPADs device, 1:10 PDMS pre-polymer was cast on the template, degassed under vacuum for 20 min, and cured at 110 °C for 20 h and gently peeled off the template on a 25 mm diameter #1 circular coverslip (Electron Microscopy Services). The peeling-induced collapse of the mPADs was rectified by sonication in 100% ethanol, followed by supercritical drying in liquid $CO_2$ using a critical point dryer (Samdri-PVT-3D; Tousimis). Flat PDMS stamps were generated by casting 1:20 PDMS pre-polymer on flat and silanized silicon wafers. Stamps were coated in a saturating concentration of fibronectin (Thermo Fisher D307) (50 µg/ml in PBS) and AF647-fibrinogen (Thermo F35200, 20 µg/mL) for 1 h. These stamps were washed in sterile distilled water and dried under a stream of nitrogen gas. Subsequently, fibronectin-coated stamps were placed in contact with surface-oxidized mPADs (UVO-Model 342; Jelight). mPADs were subsequently transferred to a solution of 0.2% Pluronics F127 (Sigma–Aldrich) for 30 min to prevent nonspecific protein adsorption.

Cells were seeded in a growth medium and then allowed to spread overnight. On the following day, mPAD substrates were transferred to an aluminum coverslip holder (Attoflour Cell Chamber; Invitrogen) for live-cell microscopy and placed in a stage incubator that regulated temperature, humidity, and $CO_2$ (Tokai Hit). In some experiments, adherent cells were treated with Y-27632 (10 µM) or PF-228 (1 µM) for 90 min prior to imaging. For PP2 treatment, cells were allowed to attach for 3 h and then incubated with PP2 (10 µM) or DMSO for 20 h prior to analysis.

For staining of FAs, cells on mPADs in the imaging chamber were fixed in a warm mixture of 50% cytoskeleton-stabilizing buffer (CSK buffer, pH 7.0: 0.5% Triton X-100, 10 mM PIPES buffer, 50 mM NaCl, 150 mM sucrose, 3 mM $MgCl_2$, 1 µg/mL leupeptin, 1 µg/mL aprotinin, Halt™ phosphatase inhibitor cocktail [ThermoFisher, 1:400 dilution]) and 50% PBS with 10% paraformaldehyde for 10 min at 37 °C. The cell was then permeabilized with CSK buffer with 0.5% Triton-X100 for 5 min, incubated in 0.1 M glycine solution to quench free aldehydes for 5 min, blocked in 33% goat serum in PBS for 1 h, and incubated with primary antibody against FA components overnight at 4 °C in PBS with 33% goat serum and 0.02% Tween-20. On the following day, samples were incubated with secondary antibody in PBS with 33% goat serum and 0.02% Tween-20 at room temperature for 1 h. The following antibodies were used rabbit polyclonal antibody against pY397-FAK (Abcam ab39967, 1:300), mouse polyclonal against tFAK (Millipore 4.47, 1:200).

Confocal microscopy images of FAs were taken with a Nikon C2 module connected to a Nikon Eclipse Ti inverted microscope with a high magnification objective (×60 Plan Apo VC Water Immersion objective, NA 1.2, Nikon). mPADs post images were captured using a 640 nm laser with a 685/50 nm filter, vinculin images were captured using a 488 nm laser and 525/50 nm filter, tFAK images were captured using a 561 nm laser and 595/50 nm filter, and pY397-FAK images were captured with a 405 nm laser and 450/40 nm filter.

For force measurements, the top of the posts was labeled with AF647-fibrinogen and was sequentially imaged and the deflection measured using custom MATLAB code. The resulting force, $F$, was calculated using Euler-Bernoulli beam theory, in which $E$, $D$, $L$, and $\delta$ are Young's modulus, post diameter, post height, and post deflection, respectively:

$$F = \delta \frac{3\pi E D^4}{64 L^3} \quad (1)$$

Micropillar array and characteristics dimensions:

5 kPa: Post height (8.3 µm), Spring constant (7.22 nN/µm), post diameter (1.83 µm)

14 kPa: Post height (6.1 µm), Spring constant (18.17 nN/µm), post diameter (1.83 µm)

25 kPa: Post height (5.0 µm), Spring constant (33.03 nN/µm), post diameter (1.83 µm)

Cell spreading and FA size and intensity were quantified using Nikon NIS-Elements Analysis Software and ImageJ. Heatmaps of immunofluorescence staining were generated via MATLAB with a 2-D Gaussian smoothing kernel with a standard deviation of 3 and plotted via the MATLAB 2-D contour plot function.

**Western blotting**. Cells were washed with ice-cold PBS containing calcium and magnesium and lysed in cold radioimmunoprecipitation assay buffer [(1% Triton X-100, 1% sodium deoxycholate, 0.1% SDS, 150 mM NaCl, 150 mM Tris-HCl (pH 7.2)] supplemented with Protease Inhibitor Cocktail (ThermoFisher) and Phosphatase Inhibitor Cocktail (ThermoFisher) for 20 min. Lysates were sonicated briefly and then clarified by centrifugation at $14,000 \times g$ for 15 min. Protein concentration was then determined using a BCA protein assay kit (Pierce). Equal amounts of protein (10 µg) were boiled (95 °C) in Laemmli sample buffer (2% SDS, 10% glycerol, 100 mM DTT, 60 mM Tris·HCl (pH 6.8), and 0.001% bromophenol blue) for 5 min and separated by SDS-PAGE. Proteins were transferred by electrophoresis onto 0.45 µm PVDF membranes and blocked with Tris-buffered saline (TBS) LI-COR Blocking Buffer for 1 h at room temperature. Membranes were incubated with appropriate antibodies in LI-COR Blocking Buffer supplemented with 0.1% Tween-20 (TBS-T) overnight at 4 °C. Membranes were washed in TBS-T (3 washes, 5 min each) and incubated in near-infrared conjugated-secondary antibodies (LI-COR Biosciences) for 45 min followed by 3 × 5 min washing in TBS-T. Membranes were imaged with an LI-COR Odyssey Imager (LI-COR Biosciences). The following primary antibodies were used: mouse monoclonal antibody against talin (Sigma, 8d4, 1:200), rabbit polyclonal antibody against talin-1 (Abcam ab71333, 1:1000), mouse monoclonal against vinculin (Sigma, V284, 1:8000), mouse monoclonal against FAK (EMD Millipore, 4.47, 1:500), mouse monoclonal antibody against GAPDH (Abcam, ab824, 1:1000), rabbit monoclonal antibody against β-actin (Cell Signaling, D6A8, 1:100), and rabbit polyclonal against pY397-FAK (Cell Signaling, 3283S, 1:250). LI-COR IRDye secondary antibodies (IRDye 800CW, IRDye 680RD) were used for dual-color imaging. Suppl. Fig. 14 presents the uncropped blots shown in Fig. 4 and Suppl. Fig. 7.

**Magnetic bead experiments**. The isolation of cell fractions enriched for FA complexes was based on the protocol of Plopper and Ingber[40]. Fibronectin was coupled to tosyl-activated paramagnetic Dynabeads M-450 (4.5-mm diameter; Invitrogen) as described in the manufacturer's protocol. Fibronectin-coated beads were incubated with 0.125 mM bis(sulfo-succinimidyl) suberate cross-linker (ThermoFisher) for 15 min at 25 °C. Fibronectin-coated beads were incubated with cells in DMEM containing 25 mM HEPES, glucose (4.5 mg/mL), and L-glutamine (Invitrogen) supplemented with 0.2% (w/v) bovine serum albumin and 0.4 mM $MnCl_2$ for 40 min at 37 °C. A ceramic permanent magnet was used to generate perpendicular, tensile forces on beads attached to the dorsal surface of cells. For all experiments, the pole face was parallel with the surface at 6 mm from the culture dish surface. At this distance, the force on a single bead was 30 pN for magnet DZ08-N52 (3-inch diameter × ½ inch thick, K&J Magnetics) or 10 pN for magnet DZ04-N42 (3-inch diameter × ¼ inch thick, K&J Magnetics). Force was applied for 10 min, and then DTBP cross-linker (ThermoFisher) in DMEM-HEPES was added to cells to a final concentration of 3 mM and samples were incubated for a further 30 min at 37 °C. Cross-linker was quenched with 20 mM Tris-HCl. The bound adhesion complexes were isolated in ice-cold lysis buffer (20 mM Tris-HCl (pH 7.6), 150 mM NaCl, 1% NP-40, 2 mM $MgCl_2$, 20 µg ml[−1] aprotinin, 1 µg ml[−1] leupeptin, 1 µg ml[−1] pepstatin, 0.5 mM 4-(2-aminoethyl) benzenesulfonylfluoride hydrochloride and 2 mM $Na_3VO_4$). The lysate was subjected to 30-40 s sonication (VibraCell VCX 500, Sonics & Materials). Beads were isolated from the lysate using a magnetic separation stand and washed with cold lysis buffer at least three times, and proteins were eluted from beads with reducing sample buffer (50 mM Tris-HCl (pH 6.8), 10% (w/v) glycerol, 4% (w/v) SDS, 0.004% (w/v) bromophenol blue, and 8% (v/v) β-mercaptoethanol. Protein samples were separated from beads with a magnet, resolved by SDS–polyacrylamide gel electrophoresis (PAGE), and analyzed by Western blotting using an LI-COR Odyssey and Image studio 5.0 Software (Li-COR).

**Microtissue fabrication and experiments**. Layers of SU-8 photoresist (Microchem) were patterned onto silicon wafers by the successive spin coat, alignment, UV exposure, and baking steps[30]. All masters were developed in a single step in

propylene glycol methyl ether acetate (Sigma) followed by hard bake. Poly-dimethylsiloxane (PDMS, Sylgard 184, Dow-Corning) microtissue substrates were molded from the SU-8 masters. Before cell seeding, the PDMS templates were sterilized in 70% ethanol followed by ultraviolet sterilization for 15 min before treatment with 0.2% Pluronics-F127 (Sigma) solution for 10 min at room temperature.

Cells ($1.2 \times 10^6$) were suspended in 1.75 mg mL$^{-1}$ neutralized rat tail collagen type I (Ibidi) and seeded in the device. The entire assembly was centrifuged to drive cells into the chambers. Excess solution was removed, leaving solution only within the chambers, and the remaining constructs were centrifuged once again in an inverted configuration to resuspend the cells into the collagen matrix before polymerization. A few hours after polymerization, we observed the spontaneous contraction of the collagen matrix by cells. Cantilevers incorporated within each chamber spatially restricted the contraction of the collagen matrix, whereby the contracting gels slide up the cantilevers and are then caught by the larger end caps, resulting in a large array of microtissues anchored to the tips of the cantilevers.

After 24 h culture at 37 °C, cells contracted the collagen matrix around the engineered cantilever pillars and fully formed microtissues. Full-thickness incisional wounds were generated using a dissecting knife (#11). The microtissues were cut layer-by-layer increasing depths using the visual of the microscope as a guide. Algorithms to measure the wound area and the size of the tissues from time-lapse videos were implemented in MATLAB. The program accepts an input video and based on a brightness threshold generates regions that fill empty spaces inside and around the microtissues. These areas were analyzed to calculate gap area, tissue width, gap shape, and tissue shape.

Contractile force measurements were performed using an additional step of embedding fluorescent microbeads (Fluoresbrite 17147, Polysciences) into the cantilevers to accommodate computerized cantilever deflection tracking. Fluorescent microbeads (Fluoresbrite 17147, Polysciences), embedded in the caps of the cantilevers, were used for computerized deflection tracking. Briefly, the position of fluorescent beads located at the top surface of the cantilevers was measured during the course of the experiment. After recording the time-lapse data, tissues were disintegrated with collagenase, to determine the baseline position of the same beads. Cantilever displacements were measured by subtracting the baseline position of fluorescent beads from the position at a given time point. Only tissues that were uniformly anchored to the tips of the cantilevers were included in the analysis. The displacement of fluorescent microbeads at the top of the cantilevers was then tracked using the SpotTracker plug-in. The contractile force was calculated by multiplying the sum of cantilever displacement with the spring constant ($k = 9.72$ μN μm$^{-1}$) of the cantilevers at a PDMS–curing agents ratio of 1:10 ($E = 2$ MPa). The spring constant of cantilevers was calculated using COMSOL 5.4.

For cell tracking analyses, nuclei of cells in microtissues were labeled with Hoechst 33342 (Sigma). After 24 h, microtissues were wounded and fluorescence and brightfield images were captured every 30 min for an additional 24 h with a Plan Apo ×10 (N.A. 0.45) objective mounted on a Nikon Ti Eclipse connected to a Nikon C2+ confocal microscope using a live cell incubator (37 °C, 5% CO$_2$). Cell motility package from Nikon NIS Elements AR (v4.22) software was used to automatically track individual nucleus and obtain cell trajectories. The cell trajectories were uploaded into MotilityLab (http://2ptrack.net/plots.php) and analyzed for cell speed, mean square displacement (MSD), and straightness. For fitting MSD (0–12 h) to the Persistent Random Walk Model to extract Diffusivity (D) and Directional Persistence (P), ordinary nonlinear least-squares regression analysis was used as described[41].

**Statistics and reproducibility.** All data are reported as mean ± standard deviation (SD). Normally distributed data that presented equal variances were analyzed via ANOVA (unpaired $t$-tests for 2 groups) or ANOVA with Welch's correction if the variances were not equal. Other data were analyzed using the Kruskal-Wallis non-parametric test (Mann–Whitney test for 2 groups). Dunn's tests were used for post-hoc pairwise comparisons. Linear regression analyses were performed on pooled data from FAs from multiple cells. $R^2$ values are on average >0.50. To assess the suitability of linear vs. sigmoidal fits for force-FAK data, curve fits were compared using Akaine's method, which takes into account the goodness-of-fit and the number of parameters in the model. This analysis estimated the probability that the linear fit correctly describes the data at >90% (the sigmoidal fit <10%), supporting the use of linear regression fits for the data. All analyses were performed using Prism (GraphPad Software) and a $p$-value < 0.05 was considered significant.

**Kinetic model.** A kinetic model was developed to simulate talin-FAK interactions under force. In this model, talin at FAs undergoes a reversible structural change from an unstretched to a stretched form under force. The stretched talin state reversibly binds FAK to form a FAK-talin complex. FAK in the FAK-talin complex then undergoes reversible phosphorylation. The law of mass action was applied to derive a system of coupled differential equations (Eqs. 1–3) describing the time-dependent changes for the number of stretched talin molecules ($T_{st}$), FAK-talin complexes ($FT_{st}$), and phosphorylated FAK-talin complexes ($pFT_{st}$). Conservation laws for talin and FAK numbers (Eqs. 4, 5) were applied to reduce the number of unknown variables to match the number of equations. The effect of force on talin conformation was incorporated into the kinetic rate-controlling talin conversion

into the stretched state using the Bell model[31] and in agreement with experimental measurements of talin stretching under force[13] (Eq. 6).

Governing equations

$$\frac{d}{dt}T_{st} = k_1 T - k_2 T_{st} - k_3 T_{st} \cdot F + k_4 FT_{st} \qquad (2)$$

$$\frac{d}{dt}FT_{st} = k_3 T_{st} \cdot F - k_4 FT_{st} - k_5 FT_{st} + k_6 pFT_{st} \qquad (3)$$

$$\frac{d}{dt}pFT_{st} = k_5 FT_{st} - k_6 pFT_{st} \qquad (4)$$

Conservation laws

$$T_{total} = T + T_{st} + FT_{st} + pFT_{st} \qquad (5)$$

$$F_{total} = F + FT_{st} + pFT_{st} \qquad (6)$$

Force-kinetics coupling

$$k_1 = k_{1o} e^{\alpha f} \qquad (7)$$

The system of differential equations at steady state (d/dt = 0) was solved using the SOLVE function in MATLAB (ver R2019b). Supplementary Table 1 lists the parameters used in the base model. Solutions were exported to Excel and GraphPad for plotting.

**Reporting summary.** Further information on research design is available in the Nature Research Reporting Summary linked to this article.

## Data availability
Data supporting the findings of this manuscript are available from the corresponding author upon reasonable request. A reporting summary for this article is available as a Supplementary Information file.

## Code availability
MATLAB code for the kinetic model is available at https://github.com/MarcFdz/Force-FAK-coupling.

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

## Acknowledgements

The authors thank B. Koehler for help with data analysis and T.H. Barker for recommendations related to magnetic bead assay. We acknowledge support from the National Institutes of Health Award No. R01 EB024322 (A.J.G.), R01 CA254342 and R01 CA247562 (D.D.S.), and R21 EB028491 (J.E.), and the National Science Foundation Graduate Research Fellowship (D.W.Z.) under Grant No. DGE-1148903 and the NSF Science and Technology Center for Engineering Mechanobiology (C.S.C.). The contents of this publication are solely the responsibility of the authors and do not necessarily represent the official views of the NIH or NSF.

## Author contributions

D.W.Z., M.A.F.-Y., N.S.C., E.N.H., E.B.O., A.J.G. conducted experiments and analyses; A.F.G. wrote and validated MATLAB code for the model. A.J.G. conceptualized and supervised the research. D.D.S. provided the mutant FAK cell lines and input on interpretation of results. J.F. provided the mPADs master molds and suggestions on experimental design and data interpretation. J.E. and C.S.C. provided masters for microtissue molds and input on interpretation of results. D.W.Z., M.A.F.-Y., and A.J.G. wrote the manuscript, with contributions from all other authors.

## Competing interests

The authors declare no competing interests.
