## [Peer Review File · Nature Communications]

Reviewer #4 (Remarks to the Author):

The study of Zhou et al., examines the force-dependent dynamics of focal adhesion kinase at single adhesions using a multiplicity of elegant methods including microfabricated post array detector (mPAD) system and magnetic microbeads. They find a linear correlation with force in the presence of talin when stiff, but not soft, are pulled on via myosin II molecular motors acting through F-actin. The results are compelling and functional relevance to microtissue repair in vitro is demonstrated. In this review I focused primarily on the magnetic bead assays and the kinetic modeling that have been added to the study. Overall, these concerns are relatively minor, and I am very supportive of publication but suggest some minor revisions to improve the final version.

Magnetic bead experiments

1. Magnetic bead experiments are really nice. But is the stiffness of the magnetic force puller varying too such that the spring constant is also changing? I suggest adding a calculation to estimate the magnetic puller stiffness (spring constants) for each of the force levels. These can be directly compared to the force-displacement relationships (spring constants) of the microposts. I suspect that the bead spring constants are very low compared to the posts, and therefore the FAK effects are likely attributable to force and not stiffness in the bead experiments.

Kinetic modeling

2. The model uses force as an input constant, which enables the model to account for the kinetic states of talin and its FAK-associated states. However, the force is really a dynamic variable emerging as an output from the motor-clutch system (Chan and Odde, *Science*, 2008), which provides a mechanistic explanation for the increasing force with increasing stiffness (Bangasser et al., *Nat Comm*, 2017). Previous work has shown that talin unfolding-mediated adhesion reinforcement via vinculin can be readily layered onto the base motor-clutch model and can be applied to MEFs to predict a myriad of mechanotransduction phenomena (Elosegui-Artola et al., *Nat Cell Biol*, 2016). Thus, the present model, which does not account for these underlying drivers, seems relatively narrow and less informative compared to what could have been predicted and tested had the previous model of Elosegui-Artola et al. had been used instead. By simply adding the FAK effects to the model of Elosegui-Artola, the study would have been more powerful and better framed in context of previous work. As it is, the modeling is fine, and helps explain the final interpretation of the results, but has only a modest impact on the overall strength of the study, which is substantial.

Additional comments

3. The authors chose to present the effect of k_3 , the forward binding rate for the FAK-talin complex formation. However, it seems effect could just as well be due to the reverse rate being increased. Since affinities are usually modulated by the off-rate constant changes, this seems more plausible and should at least be mentioned.

4. In Fig. 1j, Y drug looks significant? Need to add "alpha" to figure.

5. Are the labels reversed on the Fig. 6d, g, j, i.e. blue and black labels are reversed? If not then there is a strange biphasic effect that needs an explanation.
6. The authors should mention which ode solver they used in matlab.

Response to Reviewer Comments

RE: *Nature Communications* NCOMMS-20-48775A

Reviewer #4 (Remarks to the Author):

The study of Zhou et al., examines the force-dependent dynamics of focal adhesion kinase at single adhesions using a multiplicity of elegant methods including microfabricated post array detector (mPAD) system and magnetic microbeads. They find a linear correlation with force in the presence of talin when stiff, but not soft, are pulled on via myosin II molecular motors acting through F-actin. The results are compelling and functional relevance to microtissue repair in vitro is demonstrated. In this review I focused primarily on the magnetic bead assays and the kinetic modeling that have been added to the study. Overall, these concerns are relatively minor, and I am very supportive of publication but suggest some minor revisions to improve the final version.

Response: We thank the reviewer for the very positive and constructive comments. We have addressed the reviewer's comments below.

Magnetic bead experiments

1. Magnetic bead experiments are really nice. But is the stiffness of the magnetic force puller varying too such that the spring constant is also changing? I suggest adding a calculation to estimate the magnetic puller stiffness (spring constants) for each of the force levels. These can be directly compared to the force-displacement relationships (spring constants) of the microposts. I suspect that the bead spring constants are very low compared to the posts, and therefore the FAK effects are likely attributable to force and not stiffness in the bead experiments.

Response: We thank the reviewer for recognizing the significance of the magnetic beads experiments with applied forces to examine the biological significance of the force-FAK coupling relationship. As suggested by the reviewer, we estimated the magnetic puller stiffness by dividing the bead forces by the expected range of displacements (0.3-1 μm) (Guilluy et al., *Nat Cell Biol* 2011), yielding a spring constant value in the order of 0.03 nN/ μm . This estimate is approximately 2 orders of magnitude lower than the spring constant for the mPADs (~ 7 nN/ μm). As mentioned by the reviewer, this analysis indicates that the FAK effects are likely attributable to force and not stiffness in the bead experiments. This point has been included in the revised paper.

Kinetic modeling

2. The model uses force as an input constant, which enables the model to account for the kinetic states of talin and its FAK-associated states. However, the force is really a dynamic variable emerging as an output from the motor-clutch system (Chan and Odde, *Science*, 2008), which provides a mechanistic explanation for the increasing force with increasing stiffness (Bangasser et al., *Nat Comm*, 2017). Previous work has shown that talin unfolding-mediated adhesion reinforcement via vinculin can be readily layered onto the base motor-clutch model and can be applied to MEFs to predict a myriad of mechanotransduction phenomena (Elosegui-Artola et al., *Nat Cell Biol*, 2016). Thus, the present model, which does not account for these underlying drivers, seems relatively narrow and less informative compared to what could have been predicted and tested had the previous model of Elosegui-Artola et al. had been used instead. By simply adding the FAK effects to the model of Elosegui-Artola, the study would have been more powerful and better framed in context of previous work. As it is, the modeling is fine, and helps explain the final interpretation of the results, but has only a modest impact on the overall strength of the study, which is substantial.

Response: The reviewer raises a valid point. We formulated the model with force as the input constant for direct correspondence to the metric (traction force) presented in the experimental results of the study. We have revised the manuscript to incorporate the reviewer's point in the discussion for the benefit of future analyses.

Additional comments

3. The authors chose to present the effect of k_3 , the forward binding rate for the FAK-talin complex formation. However, it seems effect could just as well be due to the reverse rate being increased. Since affinities are usually modulated by the off-rate constant changes, this seems more plausible and should at least be mentioned.

Response: We appreciate this valid comment. We conducted additional parametric simulations for the reverse rate for the FAK-talin complex formation (k_4) and include them in a new supplementary figure (Supplementary Fig. 13). As expected, increasing k_4 has equivalent effects as decreasing k_3 . This point is discussed in the revised text.

4. In Fig. 1j, Y drug looks significant? Need to add "alpha" to figure.

Response: The figure has been corrected to include the 'alpha'.

5. Are the labels reversed on the Fig. 6d, g, j, i.e. blue and black labels are reversed? If not then there is a strange biphasic effect that needs an explanation.

Response: The labels were indeed reversed, and it has been corrected.

6. The authors should mention which ode solver they used in matlab.

Response: As mentioned in the methods, we solved the system at steady state ($d/dt = 0$) which reduces the system to algebraic system which was evaluated with the 'solve' function.

Reviewer #4 (Remarks to the Author):

The authors have addressed my comments, and are to be commended on an excellent contribution.